# The Effect of Heat Treatment toward Glycerol-Based, Photocurable Polymeric Scaffold: Mechanical, Degradation and Biocompatibility

**DOI:** 10.3390/polym13121960

**Published:** 2021-06-14

**Authors:** Wai-Sam Ao-Ieong, Shin-Tian Chien, Wei-Cheng Jiang, Shaw-Fang Yet, Jane Wang

**Affiliations:** 1Department of Chemical Engineering, National Tsing Hua University, Hsinchu 30013, Taiwan; namelessandsam@yahoo.com.tw (W.-S.A.-I.); peter.j134@gmail.com (S.-T.C.); 2Institute of Cellular and System Medicine, National Health Research Institutes, Zhunan 35053, Taiwan; wcjiang@nhri.org.tw (W.-C.J.); syet@nhri.org.tw (S.-F.Y.)

**Keywords:** poly(glycerol sebacate) acrylate, thermal treatment, 3D printing, photo crosslinking, mechanical properties, biocompatibility

## Abstract

Photocurable polymers have become increasingly important for their quick prototyping and high accuracy when used in three dimensional (3D) printing. However, some of the common photocurable polymers are known to be brittle, cytotoxic and present low impact resistance, all of which limit their applications in medicine. In this study, thermal treatment was studied for its effect and potential applications on the mechanical properties, degradability and biocompatibility of glycerol-based photocurable polymers, poly(glycerol sebacate) acrylate (PGSA). In addition to the slight increase in elongation at break, a two-fold increase in both Young’s modulus and ultimate tensile strength were also observed after thermal treatment for the production of thermally treated PGSA (tPGSA). Moreover, the degradation rate of tPGSA significantly decreased due to the increase in crosslinking density in thermal treatment. The significant increase in cell viability and metabolic activity on both flat films and 3D-printed scaffolds via digital light processing-additive manufacturing (DLP-AM) demonstrated high in vitro biocompatibility of tPGSA. The histological studies and immune staining indicated that tPGSA elicited minimum immune responses. In addition, while many scaffolds suffer from instability through sterilization processes, it was proven that once glycerol-based polymers have been treated thermally, the influence of autoclaving the scaffolds were minimized. Therefore, thermal treatment is considered an effective method for the overall enhancement and stabilization of photocurable glycerol-based polymeric scaffolds in medicine-related applications.

## 1. Introduction

In the case of photopolymerization, ultraviolet (UV) light is often applied to trigger radical formation to enable the facile crosslinking of polymer chains. Radical polymerization via photocuring offers many advantages such as short curing time, low energy and temperature requirements [1,2]. With these advantages, photocuring has become a popular method in industrial manufacturing, such as coating and additive manufacturing [3,4]. With rapid innovations in materials, several existing biodegradable and biocompatible polymers have been chemically modified to convert their hydroxyl groups to acrylate groups in order to crosslink via radical polymerization, for example, photocurable chitosan [5], poly(ethylene glycol) diacrylate (PEGDA) [6], poly(capro-lactone) diacrylate (PCLDA) [7] and poly(capro-lactone) triacrylate (PCLTA) [8]. However, the disadvantages of these photocurable polymers are the poor impact resistance and brittleness due to the relatively high stiffness and low elasticity, which greatly limit their applications [9].

Biodegradable, photocurable polymers had become rather useful in soft tissue engineering, such as cardiac regeneration [10] and nerve regeneration [11] in recent year. Since many tissue and cellular functions are dependent upon the mechanical properties of the tissues, a biomimicking scaffold will include mimicking the mechanical properties of the targeted tissue [12,13]. The mechanical properties of many soft materials for tissue engineering still differ from the host soft tissue [14,15,16] at a kilopascal to megapascal scale range. However, in many of the studies, it was mentioned that the biodegradable polymers used were not strong enough. Since very little work had been accomplished on the enhancement of mechanical properties of softer biodegradable polymers thus far, it is the goals of this study to addresses the demand for soft tissue engineering.

To improve the mechanical properties of photocurable polymers, a lot of work was carried out on the modification of material formulations [17,18]. Many modifications have been made to PEGDA for mechanical improvements, such as (1) the use of additives as a filler prior to polymerization [19], (2) the utilization of copolymer systems [20,21] and (3) the introduction of other photo-polymerization mechanisms to control the arrangement of polymer chains [22]. However, most of the chemical modifications centered on the modification of pre-polymers before the formation of cross-linkages, which may ultimately change the viscosity of the pre-polymer, the overall chemical components and the biocompatibility. This greatly limits the application of photocurable polymers toward drug delivery and tissue engineering [23,24,25].

In addition to chemical modifications, modification of the crosslinked polymer after radical polymerization is another common method to improve mechanical properties. This method is often referred to as “post-treatment,” to clarify that it takes place after radical polymerization through light. Previous studies on post-treatment methods mostly focused on ultraviolet light and thermal curing of the materials. These treatments are often applied to enhance the mechanical strength of the materials by increasing the number of linkages, and to subsequently strengthen the polymer networks [26,27,28,29,30]. For many commercial acrylate resins, a secondary UV curing after the initial photocrosslinking is an effective process to improve mechanical properties of photocurable polymers without the addition of chemicals or other impurities [31,32]. Secondary UV curing is widely applied in the manufacturing of commercialized products for its simplicity and low cost and has been very well studied [18,33,34]. However, as more 3D printing companies turn toward heat treatment, which is more commonly known as “annealing”, of linear polymers, the effect toward biodegradable network polymers, such as glycerol-based polymers, are rarely studied. Most of the works on heat treatment were focus on the stiff material, such as dental composites [35]. Through this study, post-treatment process would be able to produce photocurable biodegradable polymers with compatible mechanical properties and further expand the selection of materials.

Poly(glycerol-co-sebacate) acrylate (PGSA) was developed in 2007 as a photocurable, biocompatible and biodegradable glycerol-based elastomer [36]. Photo-crosslinked PGSA formed a network polymer that demonstrated tunable mechanical properties and degradability, which is proportional to its degree of acrylation. Traditionally, it was rather difficult to form 3D structures with the thermosetting PGS, as it will melt upon curing. To make complicated 3D structures, instead PGSA was chosen and scaffolds were created through DLP-AM [37,38,39]. However, even though the mechanical property of PGSA may be tuned through different degrees of acrylation, too many crosslinkages formed through radical polymerization may eventually render the material non-degradable. Therefore, it is important to seek a balance between mechanical strength and degradability. Hence, it was proposed by this work to thermally treat 3D-printed PGSA scaffolds to enhancement the mechanical properties by increasing the number of crosslinkages while maintaining its degradability. Therefore, PGSA scaffolds with greater robustness, tunability and precision would further expand its applicability in tissue engineering and medical devices [40]. In this paper, thermal treatment with different temperatures is applied to enhance the properties of PGSA photocured scaffolds. The effect of thermal treatment and second UV curing treatment on the mechanical properties and degradability of PGSA are investigated. Moreover, the biocompatibility both in vitro and in vivo after treatment is studied to confirm the changes in terms of cell viability, proliferation rate, metabolic activity and immune responses after thermal treatment. Through the development of PGSA post-treatment method in this study, a reliable and effective treatment for the modification and enhancement of tissue engineering scaffolds is demonstrated.

## 2. Materials and Methods

### 2.1. Synthesis and Characterization of PGSA

All chemicals were purchased from Sigma-Aldrich (Taiwan), unless stated otherwise. poly(glycerol sebacate) acrylate (PGSA) was synthesized after poly(glycerol sebacate) (PGS) pre-polymer was prepared. PGS pre-polymer was synthesized through polycondensation of equimolar glycerol and sebacic acid at 130 °C under nitrogen for 2 h, then the reaction is continued under vacuum for an additional 24 h. Acrylation of PGS pre-polymer was carried out by dissolving 30 g PGS, 30 mg 4-Dimethylaminopyridine in 300 mL dichloromethane (DCM) inside a 2-neck round bottom flask under nitrogen. The flask was then cooled in an ice bath before slowly adding acryloyl chloride (0.6 mol/mol of hydroxyl groups on PGS pre-polymer) and triethylamine (equimolar to acryloyl chloride). After the addition of reactants, the solution was stirred (hot plate, PC-420D CORNING,) at room temperature for 24 h. A rotary evaporator (Younme, Taiwan) (water bath at 35 °C) was used to remove dichloromethane. Ethyl acetate was added to the PGSA pre-polymer solution to crash out the triethylamine hydrochloride. Triethylamine hydrochloride is removed by vacuum filtration, yielding a clear solution that is then extracted three times with 50 mM of hydrochloric acid. The organic layer was removed via rotary evaporator such that only the pre-polymer remained, which was then stored at 4 °C until its usage. ^1^H nuclear magnetic resonance spectra of PGS pre-polymer and PGSA pre-polymer were recorded using NMR spectrometer (VARIAN VNMRS-700, USA, 700MHz) after the pre-polymers were dissolved in DMSO-d6.

### 2.2. Network Formation and 3D Printing

PGSA pre-polymer was directly mixed with 1 wt% of Diphenyl(2,4,6-trimethyl- benzoyl) phosphine oxide (TPO) under stirring. The mixture was then cured by exposure to ultraviolet light (LED box with wavelength 385 nm, ~33 mW/cm^2^) for 30 s of each side. Photocured PGSA films were further exposed under UV light for varying durations as post UV-curing treatment. PGSAprin films were allowed for thermal post-curing treatment in a vacuum oven (DOV30, Deng Yng, Taiwan) with different temperature (120, 140, 160 °C) for 10 h.

3D printed PGSA scaffold was designed using SolidWorks™ software (2012, Dassault Systèmes SE, French) and fabricated via DLP-AM system with a wavelength of 405 nm (built by National Taiwan University of Science and Technology, NTUST). The 3D structure was separated into several layers during 3D printing and the curing time was 8 s for each layer with an operating power of the system at 16 mW/cm^2^.

### 2.3. Mechanical Measurement of PGSA via Tensile Test

The tensile test was referred to the standard of ASTM D412. Samples were cut into dog-bone-shaped polymer strips with the dimension of 30 × 3 × 1 mm^3^ (n ≥ 5 for each formulation). The tensile test was conducted using a tensile machine (ElectroForce^®^ 3200, TA Instrument, New Castle, DE, USA) with a 225 N load cell at a strain rate 3 mm/min. All the samples were elongated to failure. The initial Young’s modulus was calculated from the linear region of the stress-strain curve (the initial slope: 0–5% of the final strain). Ultimate tensile strength was obtained on the breaking point of the stress-strain curve. Elongation was calculated by the strain at break divide by the original length of the sample.

### 2.4. Thermal Analysis

The decomposition temperature of PGSA and PGSA that underwent thermal treatment were measured through thermogravimetric analysis (Q600, TA Instrument, USA) with a temperature increase rate of 10 °C/min from room temperature to 550 °C at normal atmospheric pressure under a gas flow rate at 100 mL/min. Moreover, the above materials were cut into dog-bone-shaped as same as the samples in tensile test for dynamic mechanical analysis (Q800, TA Instrument, USA). The temperature was increased from −70 to 50 °C at 5 °C/min to measure the glass transition temperature (Tg), also known as “tan delta”.

### 2.5. In Vitro Degradation Test

PGS (curing at 160 °C for 8 h), PGSA and PGSA that underwent thermal treatment were degraded in PBS with lipase from porcine pancreas at a concentration of 20 units/mL and hydrolyzed in 0.2M sodium hydroxide. Samples were incubated in 100% ethanol to remove sol and weighed after drying in oven at 60 °C for 24 h. The samples were incubated in 12-wall cell culture disks. Each wall was poured with 4 mL lipase solution, incubated at 37 °C and replaced by with enzyme solutions every two days. At day 6, 12, 18, 24, 30 (2 to 386 h for hydrolysis), samples were collected and rinsed with deionized water (DI water) and dried at 60 °C for 48 h before determining the mass loss. Water swelling ratios of the samples were measured through weighting before and after the degradation tests. Degradation rate and swelling ratio are calculated by comparing the percentage change of mass.

### 2.6. In Vitro Biocompatibility

Hig82 cell line was cultured in Ham’s F12 nutrient mix medium (F12) supplemented with 10% (*v*/*v*) fetal bovine serum and 1% (*v*/*v*) antibody (Pen-Strep Ampho. solution) (growth medium using in cell culture) at 37 °C and 5% CO_2_. Prior to seeding, Hig82 was allowed to proliferate in a 25T flask. The medium in 25T flask was removed and rinsed with PBS. Cells were harvested after incubating in 1 mL trypsin (0.25%)/EDTA (0.02%) for 3 min and quenched with four times volume of growth medium to suspend the cells again. The mixture was centrifuged at 1000 rpm for 3 min. The transparent phase was collected and mixed with another 8 mL of medium.

Samples were punched into circular films with a diameter of 6 mm and incubated in ethanol before seeding. Samples were placed into a Costar Polystyrene 96 well-plate with ultra-low attachment surface and were incubated in PBS for one day under UV sterilization. The PBS was removed after one day. Each well was then seeded with 0.5 × 10^4^ cells/mL using 0.2 mL of growth medium. PrestoBlue assay (ThermoFisher, Waltham, MA, USA) was performed at day 1, 3, 5 and 7 by adding 10 *v*/*v*% PB reagent to the medium and incubated for 2 h. After incubation, 0.1 mL of the growth medium mixture was transferred into a separate well. Cell viability was assessed by comparing the absorbance at 570 and 600 nm. Manual cell counting is performed with a Marienfeld counting chamber (0.0025 mm^2^) after 200 μL of trypsin solution was added to each sample and incubated at 37 °C and 5% CO_2_ for 5 min. A sample of cell suspension (20 μL, suspended by trypsin) is taken with a pipette and injected within the counting chamber. The microscope is then focused on an area of the counting chamber, and the cells are manually counted. Each well is counted with a total of four cell suspensions, and the results are averaged for a cell count.

### 2.7. Metabolic Activity

Human HepG2 cells (BCRC, Hsinchu, Taiwan) were cultured in Dulbeco’s modified Eagle medium (DMEM) supplemented with 10% (*v*/*v*) fetal bovine serum and 1% (*v*/*v*) antibiotic (Pen-Strep Ampho. solution) at 37 °C and 5% CO_2_. Prior to seeding, HepG2 was allowed to proliferate in a 25T flask. The medium was removed and rinsed the flask with PBS. Cells were harvested after incubating in 1 mL trypsin (0.25%)/EDTA (0.02%) for 3 min and quenched with four times the volume of growth medium to re-suspend the cells. The mixture was centrifuged at 1000 rpm for 3 min. The transparent phase in separated funnel was collected and mixed with 8 mL of medium.

Polymer films were punched into circular films with a diameter of 6 mm and incubated in ethanol before seeding. Samples were placed into a Costar Polystyrene 96 well-plate with ultra-low attachment surface and were incubated in PBS for one day under UV sterilization. Each well was then seeded with 8000 cells. PrestoBlue (PB) assay was performed at day 1, 4 and 7 by adding 10 *v*/*v*% PB reagent to the medium and incubated for 1.5 h. After incubation, the growth medium mixture was transferred into a separate well. The reduction value was assessed by comparing the absorbance at 570 and 600 nm. For 3D scaffold, growth medium of each well was collected at day 1 and day 4 for albumin analysis with Human Serum Albumin ELISA Kit (ThermoFisher, Waltham, MA, USA).

### 2.8. In Vivo Biocompatibility

PGS, PGSA and thermal post-treated PGSA scaffolds were cut into disks with 6 mm diameter and 0.5 mm thickness. All scaffolds were sterilized under 70% ethanol and incubated in PBS before implanting into 8-wk-old male mice (C57BL/6; National Laboratory Animal Center, Taipei, Taiwan). The animals were maintained in National Health Research Institutes (NHRI) with approval by the Institutional Animal Care and Use Committee of NHRI (#NHRI-IACUC- 108086A). Each material was implanted in the back of the mice under 2.5% avertin (250 mg/kg) anesthesia. The animals were sacrificed after 1 and 2 weeks of the implantation. The implanted polymers (n = 3 for each time point) with the surrounding tissues were harvested and immediately fixed for histology as described below. 

### 2.9. Histology and Immunofluorescence

The implants with surrounding tissue were embedded in paraffin and optimal cutting temperature compound (OCT) for hematoxylin and eosin (H&E) staining, Masson’s trichrome staining and anti-CD45 antibody staining (1:100, ARG20565, Arigo, Taiwan), respectively. The histology images of paraffin sections were record by microscope (Olympus CK-40) equipped with CCD camera (Nikon DS-Ri2). Crycosections were thawed and fixed with -20°C acetone for 10 min. After washing with PBS for 3 times, the sections were blocking with 5% fetal bovine serum (FBS) for 1 h was induced. Cells were then stained with anti-CD45 antibody, goat anti-IgG antibody (1:400, ARG21689, Arigo, secondary antibody) and 4, 6-diamidino-2-phenylindole (DAPI, VECTASHIELD^®^, Novus Biologicals, USA). The staining ratio of CD45 and DAPI are quantified by ImageJ software to compare the immune responses by randomly selecting at least nine areas (≥ 3 slides) around the implant sites.

### 2.10. Statistical Analysis

Statistical analysis was performed using f-test prior to two-tailed Student’s t-test by spreadsheet program (Microsoft Excel). Significant levels of results were set at: “*” *p* < 0.05, “**” *p* < 0.01 and “***” *p* < 0.001. All data was expressed as mean ± standard deviation.

## 3. Results and Discussion

### 3.1. Mechanical Properties of PGSA after Thermal Treatment

Thermal treatment was applied to photocured PGSA at 120 °C, 140 °C and 160 °C with vacuum to form thermal treated PGSA (tPGSA) for the improvement of mechanical properties (Figure 1a). The polymer network of PGSA formed additional crosslinks with the remaining unreacted acrylate, hydroxyl and carboxylic acid groups (Figure 1b). It is noted that PGSA was stable below 300 °C and began to crack when the temperature higher than 300 °C (Appendix A). As described by Flory et al. [41], the mechanical properties of network polymers are directly correlated to crosslinking densities. The Young’s modulus and ultimate tensile strength of tPGSA significantly increased along with the treatment temperature (Figure 2). It is evident that the high Young’s modulus and ultimate tensile strength in tPGSA demonstrated a high number of crosslinks. According to the Flory-Rhener equation (Appendix A), the ratio between the number of network chain segments of tPGSA to PGSA was 1.42 (Appendix A). Meanwhile, the average molecular mass between crosslinks (M_c_) of PGSA was assumed to be around 800 g/mol, and tPGSA was thus around 664.6 g/mol (Appendix A). Proving that after thermal treatment at 160 °C for 10 h, new ester bonds were formed due to polycondensation, leading to the two-fold and three-fold increase of Young’s modulus and ultimate tensile strength, respectively. This reaction is similar to the reaction observed in PGS film formation [37]. Conventionally when crosslinking density increases, it is expected that polymer elasticity would decrease due to chain entanglement. However, in our work, the elongation of tPGSA did not decrease upon treatment, and increased slightly. This is probably a result of chain rearrangement during the annealing process that possibly filled some microvoids, as previously described by Kocatepe et al. [42] and Cheng et al. [43], which enabled some self-healing behavior of the chains. Similar effect had previously been observed in the work of Raj et al. when a polymer was mixed with some compatibilizers [44,45]. However, none of the existing work was able to fully capture the reason behind such behavior. Due to the finite hydroxyl and carboxylic acid groups, the formation of new bonds would gradually terminate after treatment for 10 h. Considering the available functional groups in PGSA, it is clear that through thermal treatment, the stiffness of the polymer network was increased, while maintaining the elasticity.

### 3.2. In Vitro Degradation

tPGSA and PGSA were incubated in lipase solution to study the in vitro degradation. In Figure 3a, tPGSA degraded much slower than the untreated PGSA as the mass loss of each are 1% (160 °C) and 6% (120 °C) in 20 days, respectively. The degradation rate of tPGSA was observed to be slower as the treatment temperature was increased. Therefore, the mass loss of PGSA and tPGSA were primarily dominated by the amount of crosslinks in the polymer network. It is hypothesized that a greater crosslinking density would hinder the penetration of the enzyme solution into the polymer network, thus resulting in slow degradation rate. This hypothesis is supported by observing the swelling ratio of PGSA driven by the penetration of enzyme solution into the relatively hydrophobic environment inside the polymer network. In Figure 3b, the swelling ratio of tPGSA increased from 2 to 4% over 20 days of enzymatic degradation, whereas untreated PGSA increased from 2 to 10%. In addition, the increase of swelling ratio over time and the low mass loss on tPGSA indicated that more ester crosslinks are reduced to hydroxyl groups and acid groups before the degraded chains left the polymer network. This is supported by the linear mass loss of PGSA and tPGSA, indicating that surface erosion dominated as the mechanism of degradation during the 20 days. However, the increase in the swelling ratio of PGSA indicated the diffusion of enzyme solution into the polymer network. The high crosslinking density of tPGSA prevented the enzyme solution from penetrating the network and maintained a steady swelling ratio until day 20.

To further examine the degradation of PGSA and tPGSA, sodium hydroxide was employed to accelerate the hydrolysis of ester linkages. The result shown in Figure 3c demonstrated that all samples can be fully degraded over time. The degradation properties of tPGSA indicated that the temperature of treatment is a critical factor toward prolonging the degradation process. tPGSA treated at 120 °C demonstrated fast reduction within 10 h which is only slightly slower than PGSA. According to Figure 2a, new crosslinks were formed during 120 °C treatment. However, the newly formed bonds degraded almost as fast as the original bonds. Instead, when treated under elevated temperature, such as 140 and 160 °C, the hydrolysis rate had reduced significantly, and the mass loss was rather linear throughout the degradation until the end stage of degradation. The mostly linear degradation rate indicated the surface erosion property of tPGSA. However, bulk degradation also plays a role in the degradation, especially during the end stage of degradation, noted by the steep drop in mass. Overall, thermally-treated PGSA has been proven to predominantly degrade through surface erosion slowly, which is beneficial in maintaining structural stability for the majority of the polymer life.

### 3.3. In Vitro Biocompatibility

In order to understand the effect of thermal treatment on the biocompatibility of PGSA, fibroblasts were seeded to study the cell activity and viability, which were characterized through cell counting and cell viability assay (PrestoBlue assay). It was reported by Wells et al. [45] that the stiffness of the material could cause significant influence on cell adhesion and proliferation. In day 1 of Figure 4a, the difference in number of cells indicated differences in cell adhesion. tPGSA with treatment temperature at 160 °C led to 25% higher cell adhesion than PGSA. Furthermore, high proliferation rate was observed on tPGSA in day 7 where the number of cells was merely doubled on untreated PGSA. In Figure 4b, tPGSA demonstrated significant increase in PB reduction during the 7 days, confirming the higher viability and proliferation. The cell morphologies of Higs 82 on both PGSA and tPGSA over the 7 days are shown in Appendix A. Even though thermal treatment led PGSA to be relatively hydrophobic surface due to decrease of hydroxyl groups, the stiffness of material dominated the cell adhesion, proliferation rate and metabolic activity. Moreover, the thermal treatment process may help eliminating some of the cytotoxic chemicals, for example solvents, side products and photoinitiators. It is therefore concluded that in the case of PGSA, thermal treatment assists with cell adhesion and proliferation.

### 3.4. Metabolic Activity

The comparison of cell metabolic activity of HepG2 cells on PGSA and tPGSA was conducted on both flat films and 3D-printed scaffolds. After HepG2 cell were seeded for 1 day (Figure 5d), higher reduction of PB was observed on tPGSA film than PGSA film. The result suggested that thermal treatment on PGSA effectively increased HepG2 cell attachment. In addition, by comparing to PGSA, higher reduction of PB was observed on tPGSA on day 7. It is clear that cell viability of HepG2 was higher on tPGSA. With the increasing reduction of PB over time, cell viability and proliferation on tPGSA were confirmed to have increased. Thus, tPGSA is a promising material for long-term cell culture of HepG2.

The structure of 3D scaffold in Figure 5a was designed to increase the surface area for higher cell seeding efficiency. It was reported by Teng et al. [46], that high surface area and sufficient diffusion would be very critical to cell seeding efficiency and long-term cell metabolic activity. Indeed, Bokhari et al. reported that 3D culture can affect the performance of HepG2 albumin secretion [47]. In Figure 5d, both PGSA and tPGSA 3D scaffolds demonstrated higher reduction of PB than flat films. However, no significant difference in PB reduction was observed between the PGSA and tPGSA 3D scaffolds. In Figure 5e, although similar albumin concentration was observed in day 1, significant differences in albumin concentrations were found on tPGSA in day 4. HepG2 cells on tPGSA 3D scaffold demonstrated high liver functional activities. Given the same composition as PGSA but higher mechanical properties, it was proven that thermal treatment on PGSA scaffolds leads to significant increase in cell adhesion, proliferation and metabolic activity. According to the result of in vitro biocompatibility on both fibroblast and hepatocytes, thermal treatment on photocurable biodegradable PGSA after 3D printing can enhances the scaffold performance in both mechanical properties and biocompatibility for further application, such as liver fibrosis model for drug delivery study.

### 3.5. Histology

To confirm the improved in vivo biocompatibility, the host immune response was observed in mice after the subcutaneous implantation of the PGSA and tPGSA. PGS was selected as a reference group since it has been reported as a highly biocompatible biomaterials [37] and share similar degradation residues to PGSA and tPGSA. Given that PGSA and tPGSA are derivatives of PGS, in vivo biocompatibility comparable to PGS must be demonstrated before further in vivo studies. Histological sections in Figure 6 indicated the formation of fibrous capsule due to the host inflammatory response. The thickness of fibrous capsule is an indication of the degree of immune response, as collagen is synthesized by activated fibroblasts to encapsulate the foreign material [48,49]. Greater fibrous capsule thickness was observed on both PGSA and tPGSA than PGS at week 1. However, significant reductions in the thickness of fibrous capsule around PGSA and tPGSA were found at week 2. As a positive control, PGS was encapsulated by fibrous capsules of lower thickness, but also exhibited the same result of fibrous capsule thickness reduction at week 2. According to Sundback et al. [50], the thickness of fibrous caspule around PGS was similar to general healing wounds. With decreasing fibrous capsule thickness of all materials at week 2, the inflammatory response noticeably decreased, indicating that the tissues no longer find PGSA and tPGSA toxic and are clearly starting to heal.

### 3.6. Immunofluorescence

The migration of immune cells that migrated to the ECM covering the materials was observed with immunofluorescence to further examine the foreign body response to the material. The immune cells around the materials would trigger cellular responses, such as migration, proliferation, differentiation and protein synthesis, to affect the inflammation and material degradation [51,52,53]. As many immune cells, including macrophages, T cells and foreign body giant cells, can be targeted by binding to CD45 antigen, the images (Figure 7) of anti-CD45+ and DAPI staining were used to evaluate the level of immune responses. All the images were randomly captured around the implants. Weaker signals were observed around the PGSA and tPGSA films than the PGS films at week 1, followed by a significant decrease in CD45+ signal at week 2 (Figure 7c). This result indicated that acute inflammatory response to PGSA and tPGSA led to fibroblast migrations with activated macrophages and giant cells after implantation at week 1. As the healing process began, the immune cells on both PGSA and tPGSA significantly decreased. This result confirmed that the immune response of PGSA and tPGSA was similar to the highly biocompatible PGS.

### 3.7. Mechanical Properties of Different Treatment Processes

A summary of common treatments applied on PGSA are shown in Table 1. UV treatment and autoclaves are the common methods for sterilization before using in medicine. Indeed, the stability of biodegradable and photocurable polymers is required to be concerned as the treatments for sterilization would change the properties of materials. For UV treatment, it is one of the post-treatments to enhance the mechanical properties of photocurable polymers after crosslinking. The effect of UV treatment was terminated when the unreacted photoinitiators and acrylate groups were quenched to reach a maximum conversion. For crosslinked PGSA, the increase in Young’s modulus was 12% after 10 min of UV exposure. This increase in Young’s modulus was along with the exposure time, suggesting that chemical reactions took place between the unreacted acrylate groups in the polymer networks. However, a plateau in Young’s modulus was observed at 10 min, indicating the conversion was reached to maximum. To avoid radical reaction during sterilization, thermal treatment can be used before UV sterilization to stabilize the materials by eliminating the reactive functional groups.

Autoclaves are commonly applied as a post-treatment process to PGSA, to which the application of polymeric materials for tissue engineering and other medical practices often require the materials to be sterilized. In the study of Munker et al. [54], it was concluded that the high pressure heat steam in autoclaves might have played a role in inducing the deformation and exfoliation within polymeric materials, and may negatively impact their mechanical properties. In general autoclave process, it was observed that Young’s modulus of PGSA decreased about 14% and the elongation at break increased by 18%. This result suggested a possible reduction of ester groups in crosslinking networks during autoclave treatments. Even though autoclave process provides heat similar to the thermal treatment process, the former process was operated under high temperature steam while the latter was in dry environment. The steam in autoclaves facilitated the reduction of ester bonds. In comparison to PGSA, the decrease in mechanical strength of tPGSA was only 10%, indicating the stability of tPGSA in high temperature and high humidity. In the past, the sterilization through autoclaves of tissue engineering scaffolds and medical devices were unfavorable, as critical properties may change. Through the introduction of the post-treatment method, it is now possible to autoclaves most PGSA products while maintaining their mechanical integrity.

## 4. Conclusions

Photocurable PGSA demonstrated tunable mechanical properties and degradability. The application of PGSA in 3D printing may promote the fabrication of highly complex scaffolds without harsh curing conditions at the cost of biocompatibility due to the use of photoinitiators. By incorporating to thermal treatment under low pressure, more robust PGSA was created according to the dramatic increase in Young’s modulus and ultimate tensile strength with slight increase in elongation at break, while maintaining its biodegradability, though tPGSA does demonstrate longer degradation time with surface erosion properties. The study of in vitro biocompatibility illustrated the significant increase in cell adhesion, cell proliferation and liver metabolic activity of tPGSA. According to the histology and immunostaining, the result of subcutaneous implantation demonstrated a similar immune response between tPGSA and the highly biocompatible PGS. For the commercialization of PGSA products, thermal treatment poses positive effects in stabilizing the polymer networks before sterilization via either UV treatment or autoclave. Overall, this work illustrates that thermal treatment is an efficient and accessible method to enhance the mechanical properties and biocompatibility of glycerol-based photocurable polymers and demonstrates its applicability toward the manufacturing of tissue engineering scaffolds as well as biomedical devices.

## Figures and Tables

**Figure 1 polymers-13-01960-f001:**
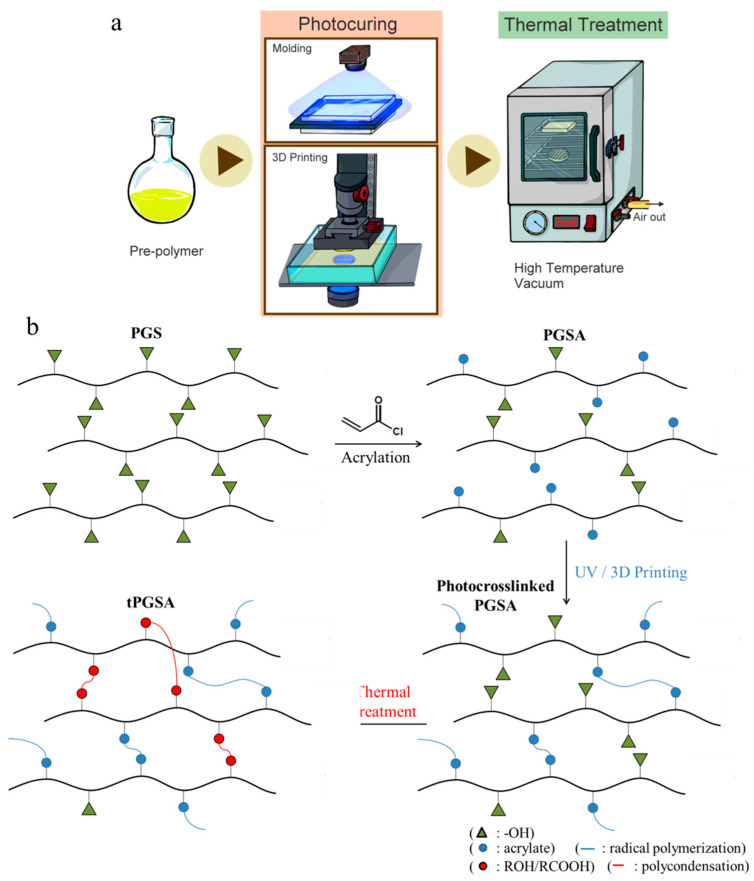
(**a**) The scheme of thermal treatment applied on PGSA after photocrosslinking. (**b**) The chemical structure and reaction of tPGSA fabricated from PGSA pre-polymer synthesis to thermal treatment, (i) PGS pre-polymer, (ii) PGSA pre-polymer (some of the hydroxyl groups were replaced by acrylate groups), (iii) PGSA network (blue line: the acrylate groups of PGSA inter-connected through radical polymerization), (iv) tPGSA network (red line: the hydroxyl groups and the carboxyl groups on PGSA polymer chains formed ester crosslinks under thermal treatment).

**Figure 2 polymers-13-01960-f002:**
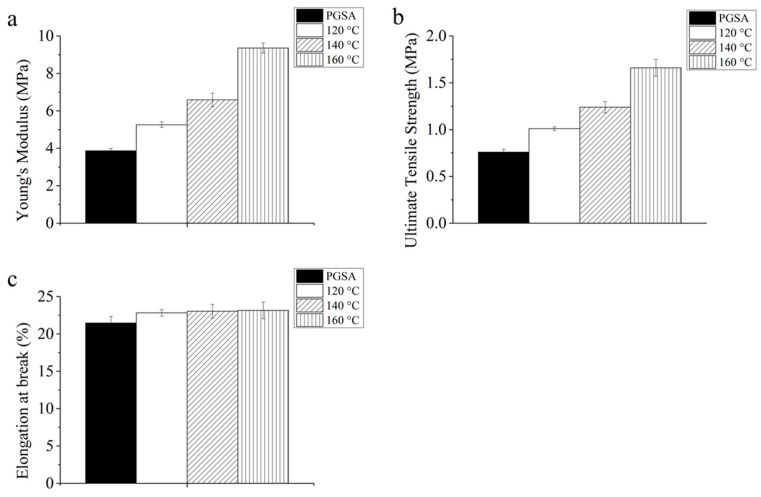
(**a**) Young’s modulus, (**b**) ultimate tensile strength and (**c**) elongation at break of PGSA and PGSA that underwent thermal treatment at 120, 140 and 160 °C.

**Figure 3 polymers-13-01960-f003:**
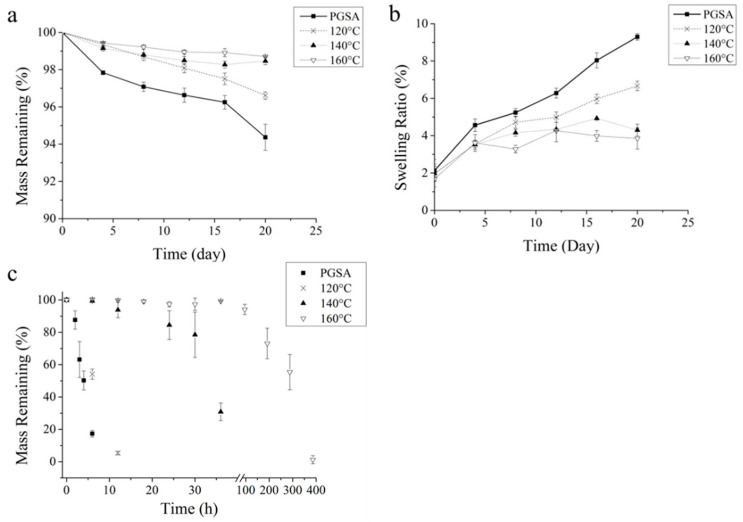
Degradation properties of PGSA and tPGSA. (**a**) Enzymatic degradation in lipase solution, (**b**) swelling ratio after enzymatic degradation, (**c**) hydrolysis in 0.2 M NaOH.

**Figure 4 polymers-13-01960-f004:**
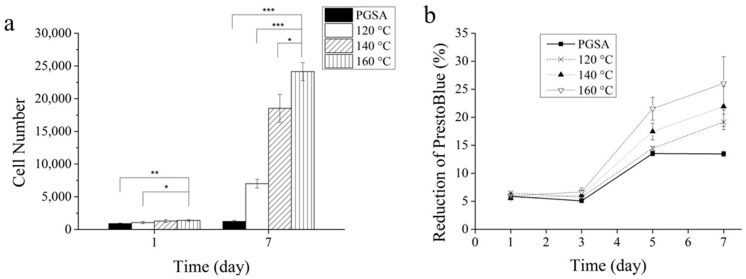
Hig82 cell line seeded on PGSA films and PGSA films undergoing thermal treatment at different temperatures and characterized by (**a**) cell counting, (**b**) PrestoBlue assay. “*” *p* < 0.05, “**” *p* < 0.01 and “***” *p* < 0.001.

**Figure 5 polymers-13-01960-f005:**
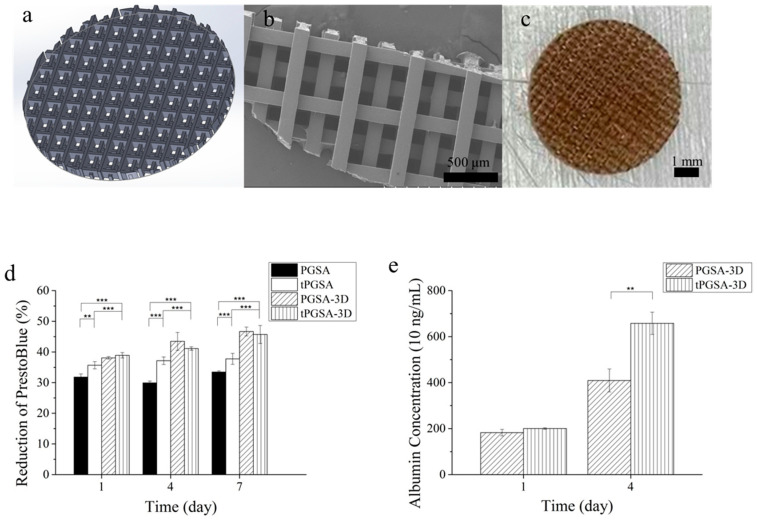
The culture of HepG2 cell line on PGSA and tPGSA. (**a**) The design, (**b**) the SEM image and (**c**) the optical image of PGSA 3D scaffold. (**d**) The cell viability of both flat films and 3D scaffolds was characterized by PrestoBlue assay. (**e**) The albumin secretion of the 3D scaffolds after culturing for 1 day and 4 days. “*” *p* < 0.05, “**” *p* < 0.01 and “***” *p* < 0.001.

**Figure 6 polymers-13-01960-f006:**
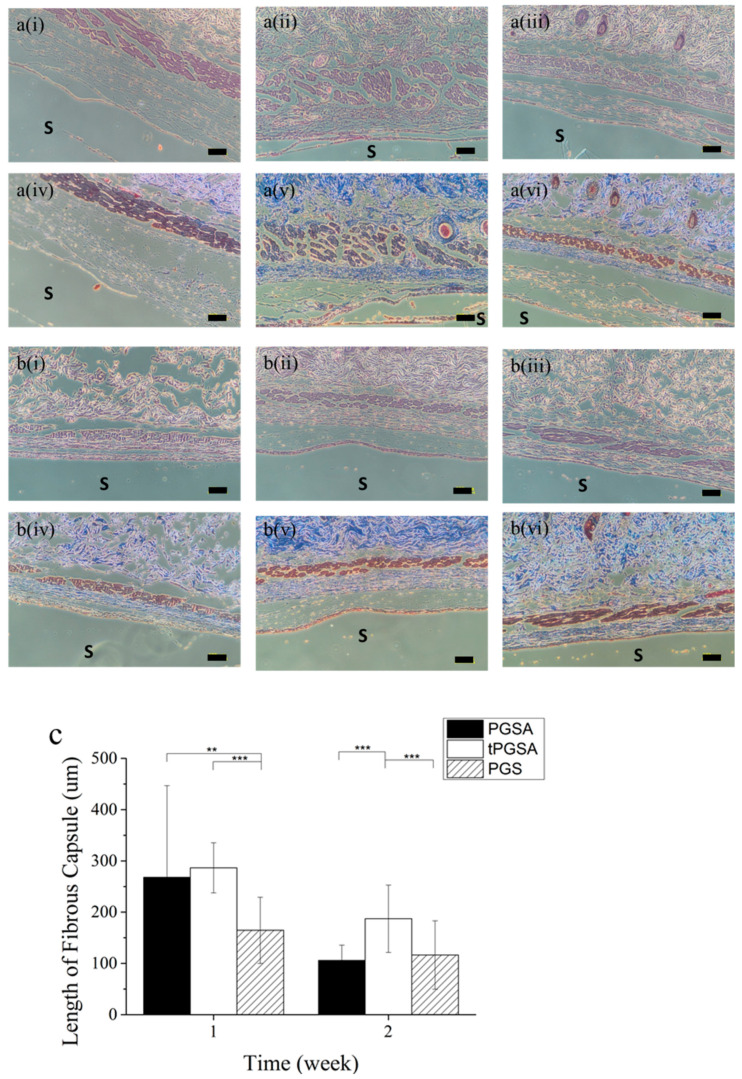
Tissue with polymers was harvested after PGSA, tPGSA and PGS have implanted subcutaneously in the mice for (**a**) 1 week and (**b**) 2 weeks. The tissue was stained with (i–iii) H&E and (iv–vi) Masson’s trichrome, which are (i,iv) PGSA, (ii,v) tPGSA and (iii,vi) PGS. (**c**) The length of fibrous capsule was evaluated using ImageJ. The site of the implanted samples was marked as “S”. Scale bar equal to 100 μm. “*” *p* < 0.05, “**” *p* < 0.01 and “***” *p* < 0.001.

**Figure 7 polymers-13-01960-f007:**
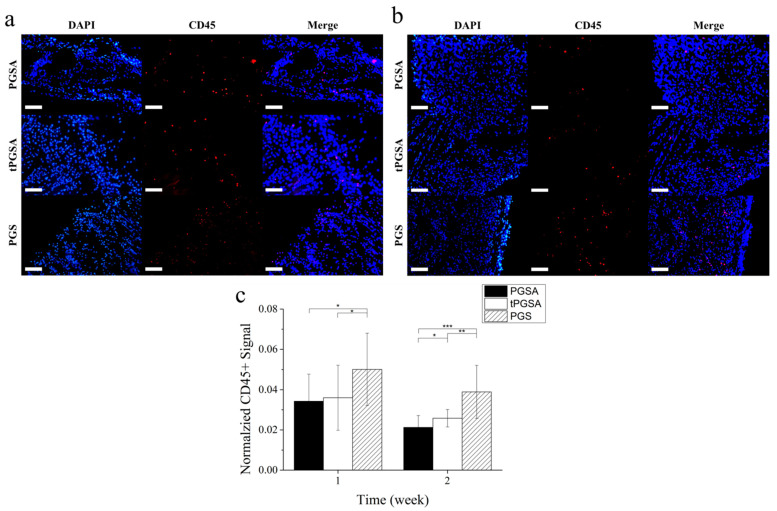
The cryosections of PGSA, tPGSA and PGS were stained with DAPI and anti-CD45+ antibody after implantation for (**a**) 1 week and (**b**) 2 weeks. (**c**) The quantitative of the staining signals was evaluated as normalized CD45+ signal. Scale bar equal to 100 μm. “*” *p* < 0.05, “**” *p* < 0.01 and “***” *p* < 0.001.

**Table 1 polymers-13-01960-t001:** Mechanical properties of PGSA underwent second UV treatment, thermal treatment and autoclave.

Treatments	Young’s Modulus (MPa)	Ultimate Tensile Strength (MPa)	Elongation (%)
PGSA	4.52 ± 0.37	0.82 ± 0.07	21.85 ± 1.84
PGSA + UV (10 min)	5.22 ± 0.30	0.84 ± 0.08	19.14 ± 1.25
PGSA + Autoclave	3.89 ± 0.15	0.81 ± 0.04	25.69 ± 0.78
PGSA + Thermal	9.36 ± 0.27	1.66 ± 0.09	23.15 ± 1.51
PGSA + Thermal + Autoclave	8.95 ± 0.40	1.47 ± 0.09	20.18 ± 1.38

## Data Availability

Not Applicable.

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
