# Peer review of "The Effect of Heat Treatment toward Glycerol-Based, Photocurable Polymeric Scaffold: Mechanical, Degradation and Biocompatibility"

_polymers, 2021, doi:10.3390/polym13121960_

Round 1

Reviewer 1 Report

In this article by Wai-Sam Ao-Ieong et al., the authors synthesized and evaluated a photocurable polymeric scaffold. The results are in line with the aims and scope of the journal. However, the article contains a number of significant shortcomings, so a major revision is requested. Please find the comments below:
1) The abstract is divided into two sections, which should be united. Besides that, the abstract itself should be made more concise as it is too long. Please enable readers to quicky familiarize themself if this article is interesting to read or not for someone.
2) Please remove the unnecessary part from the template between lines 100 and 115.
3) Please specify the frequency of the NMR in Line 134.
4) It is of utmost importance for a research paper to be reproducible. Only upon meeting this condition the community may verify the findings and build on them. The evaluated paper, however, lacks certain critical details on this front. For instance, there is no mention of the amount of PGSA pre-polymer mixed with TPO (Line 136). Moreover,  in section 2.4. the atmosphere and gas flow rate are missing. These are just examples. Please carefully screen the article and include the necessary parameters. 
4) Headlines should not be separated from the corresponding sections (Lines 147, 244). 
5) There is lots of empty space on Page 6. Please populate it to increase the readability. 
6) The schematics in Fig. 1 are too small to read.
7) Captions to figures should not be separated as well (Line 273). 
8) The figures are blurred. Please replace them with proper images. 
9) In the SEM image in Fig. 5a, the SEM micrograph is small. Please provide a proper scale bar. 
10) Please pay attention to how you represent panels of the images. There are unexpected symbols. 

Reviewer 2 Report

The current manuscript provides a non-rationalized account of thermal crosslinking profile of PGSA. On a simple google search of an undisclosed bioink manufacturer, it is clear that PGSA is a bioresorbable polymer with tunable physical properties ranging from an elastomer to a thermoset. It is reported to have great biocompatibility and excellent mechanical properties, making is useful for various tissue engineering and regenerative medicine applications. PGS is already know for its heat-induced crosslinking capabilities. So it is not clear what the authors would like to solve in the current research.

The authors mentioned that "As described by Flory et al, the mechanical properties of network polymers are directly correlated to crosslinking densities." However, the authors failed to report the crosslinking density (values), the length of the crosslinks, and the number. This becomes very important when a polymer system is 'fully' crosslinked? It is also not clear as to what makes the system so flexible even after high crosslinking?

It will be interesting to see the cell morphologies in/on the polymeric film/matrix.

Reviewer 3 Report

Dear Authors

Overall, the manuscript is interesting. I believe that it brings a lot of knowledge to the development of biocomposites. My main comments concern the methodological part. The apparatus used should be specified in more detail here. I think that in some cases it is also possible to provide more information about the course of the research, e.g. mechanical strength, TGA, spectrophotometry.

Detailed comments are provided below:

Line 95: I think that the purpose of the research (the purpose of scientific work) should be better emphasized.
Line 101 - 115: This piece of text does not apply to manyscript. It is probably copied accidentally from the example.
Line 116: I admit I prefer Materials and Methods.

Line 118: Each acronym in the text should be expanded (clarified) before first use.

Line 126: Enter the model of the mixer.
Line 134: Each test apparatus should be well described (manufacturer, country, city). All work should be reviewed in this regard.
Line 140: Enter the model of the furnace used in the tests …… .. Moreover, on what basis was the exposure time of 30s.

Further, on what basis was the curing time of 10 hours established.

Line 142: Specify the Solidworks symbol, year (Manufacturer, country, city).
Line 149: Tensile test: A test standard should be specified.

Line 149: (manufacturer, country, city).
Line 156: Add apparatus for TGA testing.
Line 236: As above (manufacturer, country, city).
Figure, 2, 3, 4, 5, 6, 7. In my opinion, there should be a larger font on the axes of the charts.
Line 457: I think one more general conclusion should be added. A forward-looking conclusion (summarizing the validity of the research).

Round 2

Reviewer 1 Report

Thank you for the corrections. I recommend the publication of the article in the present form. 

Reviewer 2 Report

Thank you for addressing all the comments. No further comments.